# Is Cell-Free DNA Testing in Hepatocellular Carcinoma Ready for Prime Time?

**DOI:** 10.3390/ijms241814231

**Published:** 2023-09-18

**Authors:** Sravan Jeepalyam, Ankur Sheel, Aslam Ejaz, Eric Miller, Ashish Manne

**Affiliations:** 1Department of Internal Medicine, University of Kansas Medical Center, Kansas City, KS 66103, USA; 2Department of Internal Medicine, The Ohio State University College of Medicine, Columbus, OH 43210, USA; 3Division of Surgical Oncology, Department of Surgery, The Ohio State University Wexner Medical Center, 320 W. 10th Ave., M-260 Starling-Loving Hall, Columbus, OH 43210, USA; 4Department of Radiation Oncology, The Arthur G. James Cancer Hospital and Richard J. Solove Research Institute, The Ohio State University Comprehensive Cancer Center, Columbus, OH 43210, USA; 5Department of Internal Medicine, Division of Medical Oncology, The Arthur G. James Cancer Hospital and Richard J. Solove Research Institute, The Ohio State University Comprehensive Cancer Center, Columbus, OH 43210, USA

**Keywords:** hepatocellular carcinoma, liver cancer, mutation, methylation, epigenetic, cell-free DNA, precision medicine, prognostic, predictive

## Abstract

Revamping the current biomarker landscape of hepatocellular carcinoma (HCC) with cell-free DNA (cfDNA) could improve overall outcomes. The use of commercially available cfDNA testing (also known as liquid biopsy) is limited by the low prevalence of targetable mutations and does not have any prognostic or predictive value. Thus, current cfDNA testing cannot be relied upon for perioperative risk stratification (POR), including early detection of recurrence, long-term surveillance, predicting outcomes, and treatment response. Prior evidence on cfDNA mutation profiling (non-specific detection or gene panel testing) suggests that it can be a reliable tool for POR and prognostication, but it still requires significant improvements. cfDNA methylation changes or epigenetic markers have not been explored extensively, but early studies have shown potential for it to be a prognostic biomarker tool. The predictive value of cfDNA (mutations and EM) to assist treatment selection (systemic therapy, immune-checkpoint inhibitor vs. tyrosine kinase inhibitor) and to monitor response to systemic and locoregional therapies should be a future area of focus. We highlighted the unmet needs in the HCC management and the current role of cfDNA testing in HCC in addressing them.

## 1. Introduction

Hepatocellular carcinoma (HCC) is the third leading cause of cancer-related death worldwide, with a 5-year survival rate of 18% [1,2]. HCC management requires a multidisciplinary approach, and multiple societies around the world have proposed guidelines for risk-stratification and appropriate treatment selection [3,4]. In early/intermediate-stage tumors, curative surgical resection (SR) and locoregional therapies (LRT) ranging from image-guided ablation, transarterial chemoembolization (TACE), stereotactic body radiotherapy (SBRT), and transarterial radioembolization (TARE) are preferred in the first line. Systemic therapy (ST) with immune-checkpoint inhibitors (ICI) and tyrosine kinase inhibitors (TKI), and vascular endothelial growth factor inhibitors (VEGFi) are often reserved for unresectable/advanced tumors. In this review, we focused on the gaps in HCC management and summarized the current status of cfDNA testing in addressing them.

### 1.1. Unmet Need in HCC Management

SR for early-stage HCC may be curative, but outcomes are less favorable for patients with aggressive features (macrovascular invasion or MVI) found on final pathology [5,6]. The 5-year survival rate is around 75% in early-stage tumors and 35% in intermediate/advanced-stage tumors (median overall survival (OS) of 37 m) [5,7]. Median progression-free survival (PFS) after surgery is around 9 months, and 38% recur in less than 2 years [8]. Post-recurrence 5-year survival rate is around 43% and current evidence does not justify the use of adjuvant (AT) or neoadjuvant (NAT) as standard-of-care (SOC). However, interim results of IMbrave 050 (atezolizumab and bevacizumab vs. surveillance in high-risk HCC) presented at the American Association for Cancer Research (AACR) this year are encouraging, with a 28% improvement in progression-free survival (PFS) (HR= 0.72, 95% CI, 0.56, 0.93; *p* = 0.0120) with adjuvant therapy. The risk stratification is based on tumor burden, MVI, and differentiation. 

LEGACY study, TRACE, and RASER trials have shown that TARE is a favorable option in uncomplicated solitary HCC in patients with Child–Pugh score (CP) A and good performance status (PS), but the failure of other prominent trials (SORAMIC, SARAH, SIRvsNIB) indicates that it may not be an ideal option over ST in the treatment of advanced tumors (CP B, high tumor burden, MVI) [9,10,11,12,13,14]. Milan’s prognostic scale (based on tumor location, MVI, total bilirubin, and tumor burden) has prognostic value based on retrospective review, but is not typically used in clinical practice [15,16]. SBRT is another local therapy for patients not eligible for resection. Recent data from RTOG 1112 showed an improvement in overall survival (OS) and PFS when SBRT was combined with sorafenib over sorafenib alone. Multiple other studies have demonstrated the utility of SBRT for the treatment of HCC [17,18,19,20,21,22,23,24,25]. 

Treatment selection (SR vs. TACE vs. SBRT vs. TARE vs. ST) is often based on the patient and tumor-related characteristics such as size, number, and location of the lesions, baseline liver functional status (by CP or Albumin-Bilirubin (ALBI) score), PS, and feasibility of the procedure intended. One of the main reasons for poor outcomes is the lack of biomarkers with a reliable prognostic or predictive value that guides HCC management. Alpha-fetoprotein (AFP) has poor sensitivity and specificity to detect recurrence or assess treatment response [26]. Similarly, imaging modalities (CT or computerized tomography and MRI or magnetic resonance imaging) lack the necessary accuracy to detect early recurrences, particularly after LRTs such as TARE or SBRT. Oftentimes, identification of residual disease or progression is delayed for months due to post-treatment changes seen early following treatment [27]. 

Genomic testing and tumor or cell-free DNA (cfDNA) do not have roles in HCC management in its current form except in identifying rare targetable mutations in advanced cancers. Circulating tumor DNA (ctDNA), genomic material from the tumor cells, is often used synonymously with cfDNA, but there is a considerable difference between them. cfDNA refers to DNA floating in the bloodstream, including ctDNA, circulating tumor cells (CTC), exosomal DNA, and DNA from normal cells [28]. The access to tissues is limited in HCC as it is generally not acquired for diagnosis, relying on distinctive imaging features. Hence, cfDNA testing is the main source of genomic testing for HCC. The use of cfDNA has the potential to address three areas of unmet need biomarker testing in HCC: (i) perioperative risk stratification (POR) to guide surgeons in identifying high-risk populations for recurrence and poor survival; (ii) prognostic biomarkers to estimate poor outcomes in intermediate/unresectable HCC; (iii) predictive biomarkers to select appropriate ST (ICI vs. TKI) or LT (SBRT vs. TACE vs. TARE) and identify the population who benefit from post-procedural ST.

### 1.2. cfDNA Landscape in HCC

Mutation profiling of cfDNA in HCC patients is commercially available and accessible for daily practice. The most frequently detected mutations are TP53 (32–80%), TERT (42–69%), CTNNB1 (17–42%), ATM (25–39%), and ARID1A (7–13%) [29,30,31,32,33,34,35,36]. Other important mutations with <10% prevalence are PIK3CA, CDKN2A, BRAF, MYC, and HER2. Concordance rates between tumor and cfDNA are reliable (52 to 83%), and especially relevantly, 27–72% of additional mutations were identified in cfDNA that were not identified in the tumor sample [31,34,37,38,39,40,41]. Multiple studies have reported high detection rates (60–100%) of non-specific cfDNA mutations, but their clinical importance is unclear [32,33,42,43,44,45]. Alternatively, epigenetic marker (EM) testing (DNA-methylation, histone modification, chromatin remodeling, and RNA-associated silencing) in HCC is still in its primitive stage, and there is increasingly encouraging evidence in other solid tumors [46,47]. This paper will focus on DNA-methylation changes and will be addressed as EM going forward. Exploring cfDNA EM is well supported by overwhelming evidence from tumor (tissue)-based studies in the literature [48,49,50,51,52,53]. 

HCC patients tend to have lower methylation levels (median: 59.3% vs. 76.9%) in cfDNA than normal controls [54]. Signatures constructed after analyzing the data from whole genome methylation sequencing (WGMS) were proposed in some studies for diagnosis and prognostication, but most prior studies tested a panel of EM [55]. There is a commercially available multianalyte test that uses a panel of more than 500 genes for methylation [56]. Smaller panels with four promoters, APC, GSTP1, RASSF1A, and SFRP1, also proved to be effective [57]. All HCC patients had at least one methylated marker from this panel. Other studies with individual markers showed good prevalence, such as LINE1 (100%), RASSF1A (87%), p16 (76%), IGFBP7 (65%), XPO4 (64%), APC (62%), CCND2 (55%), TFPI2 (47%), and DAPK (41%) [58,59,60,61,62,63]. LINE-1 is hypomethylated and the rest are hypermethylated. Concordance rates with tissue methylation profiling are reliable (68% to 89%), making it a formidable biomarker for clinical use [64]. Newer technology known as enzymatic methylation sequencing is very sensitive and requires less blood than traditional bisulfite methylation sequencing [65]. Apart from straightforward mutation or methylation testing, many cfDNA-based studies, such as DNA integrity index (DII), combinations of mutations and microRNA or circulating tumor cell and methylation, and cfDNA fragmentation patterns, are being explored to develop new biomarkers for HCC [56,66,67,68,69]. 

Etiology-specific cfDNA markers were reported in previous studies that can help in certain situations. Hepatitis B (HBV) carriers with HCC tend to have higher circulating ERBB2 and TERT mutations, higher methylation rates in RASSF1, TFPI2, TRG5 (along with AFP), and XPO4, and low methylation rates in CDKN2A than those without HCC [43,59,63,70,71,72]. Higher RASSF1 methylation rates are frequent in hepatitis C (HCV) patients with HCC (compared to HCC-negative) [73]. Some of them can help in detecting the recurrence (e.g., virus-host chimera DNA (vh-DNA), generated from junctions of HBV integration in the HCC chromosome in HBV-HCC patients, or forecasting the outcomes (e.g., higher cfDNA levels in HCV-carriers) [74,75]. The current evidence on cfDNA testing in HCC is summarized below based on the areas of unmet need discussed above. 

## 2. Perioperative Risk Stratification (POR) with cfDNA Testing in HCC

The prior cfDNA studies for POR can be broadly divided into three main categories (Table 1): (a) the detection and quantification of cfDNA levels (high vs. low); (b) detection and quantification by mutation allelic fraction (MAF) of specific mutations using small panels (hot-spots) or larger panels with next-generation sequencing (NGS) or whole exosome sequencing (WES); (c) EM. Evidence suggests that mutation profiling with larger panels (by NGS or WES) are not valuable over carefully selected hot-spot panels, and epigenetic testing needs a considerable amount of work before we can use it in clinical practice. 

cfDNA testing preoperatively (preop) helps select a population of patients who can benefit from NAT, and by serial testing, we can determine the ideal time for resection. Similarly, postoperative (post-op) serial testing can complement current SOC (AFP and imaging). The mutation profiling was shown to be more robust than imaging and other protein markers such as AFP, AFP-L3%, and des-gama-carboxy prothrombin (DCP) [76]. The ctDNA testing detected recurrence on an average of 4.6 months earlier than imaging, which is the current gold standard [77]. Patients with postoperative ctDNA-positivity were identified as significant adverse risk factors for PFS and OS. DCP-positivity combined with ctDNA was shown to increase sensitivity for detection of minimal residual disease. 

A tissue agnostic serial ctDNA mutation test proved useful to predict recurrence and microvascular invasion by serial monitoring [40]. The ctDNA detection was identified by PCR on preoperative samples, and whole exosome sequencing (WES) was performed on postoperative cfDNA samples. Serial ctDNA testing was more sensitive for poor recurrence rates than AFP or DCP. Interestingly, the ctDNA detection rate was lower (15%) than reported in other studies and there were non-synonymous mutations both in cfDNA and tumor tissue.

**Table 1 ijms-24-14231-t001:** cfDNA testing for perioperative risk stratification in hepatocellular carcinoma.

	Publication (n)	Methods	Tested Markers	Findings	Additional Information
cfDNA level	Ren et al., 2006 [78] (79) ^Preop^	UV transilluminator	Circulating DNA levels	High levels are associated with worse PFS and OS (*p* = 0.017 and 0.001, respectively)	Cut-off ≥ 36.6 ng/mL Circulating DNA level was associated with tumor size (≤5 vs. >5 cm) (*p* = 0.008) and TNM (I-II vs. III-IV) stage (*p* = 0.040). TNM-6th edition
	Tokuhisa et al., 2007 [75] (87) ^Preop^	Polymerase chain reaction (PCR)	GSTP1 gene	Patients with high cfDNA levels had shorter OS (*p* = 0.017).	Cut-off is 117.8 ng·mL^−1^ High cfDNA level to be an independent prognostic factor for OS (HR, 3.8; 95% CI, 1.7–8.5; *p* = 0.001) and cancer recurrence in distant organs (HR, 4.5; 95% CI, 1.3–14.9; *p* = 0.014). Higher cfDNA levels are associated with early intrahepatic recurrence but were not significant on logistic regression.
Mutations	Fu et al., 2022 [79] (258) ^Preop^	PCR	High-risk genes APC, ARID1A, CDKN2A, FAT1, LRP1B, MAP3K1, PREX2, TERT, and TP53	Number of mutant genes. Early recurrence (HR = 2.2, *p* < 0.001) and worse PFS in high-risk group (HR = 13.0, *p* < 0.001)	Low vs. mid vs. high-level based on the number of mutant genes. Nomogram constructed with TNM staging and risk level has accuracy with C index of 0.76 (95% CI 0.70–0.82).
	Liao et al., 2016 (41) ^Preop^ [80]	PCR	Hotspot mutations in TERT, CTNNB1, and TP53	ctDNA+ tumors have poor PFS (89 vs. 365 days, *p*< 0.001) and are at high risk of vascular invasion (*p* = 0.041)	100% specificity of recurrence in ctDNA+ patients No relationship between detectable mutations and concentration of cfDNA (*p* = 0.818).
	Ye et al., 2022 (96) ^Postop^ [81]	NGS	Hot exons of 293 genes	ctDNA-positive tumors have worse PFS and OS. ctDNA + and high-AFP have the worst PFS	Detection of even one mutation was considered ctDNA+ AFP cutoff was >400 ng/mL
	An et al., 2019 (26) ^Postop^ [82]	NGS	354 genes	ctDNA-positive patients have worse PFS (17.5 vs. 6.7 months (m), HR = 7.655, *p* < 0.0001)	ctDNA positivity is the independent risk factor for prognosis (HR = 10.293, *p* < 0.0001) The number of mutations, VAF, and ctDNA concentration correlated with tumor size.
	Zhou et al., 2020 (97) ^Postop^ [83]	NGS	1021 gene panel	ctDNA + associated with recurrence (100%), shorter PFS (5 m vs. NR, *p* < 0.001)	ctDNA+ with high AFP (>400 ng/mL) is the worst prognostic feature for recurrence.
	Cai et al., 2019 (34) ^Serial^ [76]	Targeted sequencing for SNV and WGS for CNVs	Tissue agnostic Somatic SNVs and CNVs	ctDNA+ have worse PFS and OS (*p* =0.001 for both)	Can predict recurrence on an average of 4.6 m before imaging could catch it Better than AFP, AFP-L3%, and Des-Gamma-Carboxy Prothrombin (DCP).
	Ono et al., 2015 (46) ^Serial^ [40]	ctDNA detection by PCR on preop sample. Whole exome sequencing (WES) of cfDNA in postop sample	Tissue agnostic. Whole genome sequencing (WGS) on tumor tissue	High recurrence rates and extra-hepatic metastasis ctDNA+ group (*p* = 0.01 and 0.03, respectively)	Detected in 7/46 patients ctDNA+ patients are at high risk of microscopic vascular invasion (OR 6.10; 95% CI, 1.11–33.33, *p* = 0.038) Concordance rate was 83% Included patients with hepatectomy and liver transplant, and one patient with HCC-CCA variant
	Zhu et al., 2022 (41) ^Serial^ [42]	NGS	NGS	ctDNA-positive (both pre- and post- op) patients tend have intrahepatic recurrence, and have shorter PFS Preop ctDNA-positive tumors tend be larger (>5 cm), have MVI, and higher differentiation.	Detection of NRAS, NEF2L2, and MET mutations associated with shorter PFS Higher MAF preop is an is a strong prognostic factor for PFS
Epigenetic markers	Tsutsui et al., 2010 (70) ^Preop^ [61]	PCR	CCND2 (methylated)	Patients with CCND2 gene methylation have shorter PFS (*p* = 0.02)	Cut off was >70 pg/mL serum) CCDN2 methylation is an independent risk factor for PFS (HR = 1.866, 95% CI: 1.106–3.149).
	Li et al., 2018 (155) ^Postop^ [84]	PCR	IGFBP7 methylation status	IGFBP7 promoter methylation was significantly correlated with OS (*p* < 0.001), was an independent prognostic predictor for OS (*p* = 0.000) and early tumor recurrence (*p* = 0.008), and was associated with vascular invasion (*p* = 0.014)	DNMTs mRNA levels, malondialdehyde (MDA), xanthine oxidase (XOD), glutathione hormone (GSH), and glutathione-S-transferases (GST) levels were also tested MDA and XOD levels were significantly higher in the IGFBP7 methylated group than the unmethylated group, while GSH level was lower in the methylated group than in unmethylated group (DA *p* = 0.001) (XOD *p* < *0*.001). The DNMT1 and DNMT3a mRNA levels were higher in the IGFBP7 methylated group than unmethylated group (*p* = 0.002)

Preop—preoperative sample was tested, postop- postoperative sample was tested, UV—ultraviolet. PFS—progression free survival, OS—overall survival, HR—hazard ratio. AFP—alpha fetoprotein, HCC—hepatocellular carcinoma. CCA—cholangiocarcinoma, OR—odds ratio, MAF—mutation allelic fraction.

## 3. Prognostic Value of cfDNA Testing in HCC

Reliable blood-based (cfDNA, proteins, or CTC) markers that can foretell poor clinicopathological features beyond standard imaging modalities (CT or MRI) may be helpful in treatment planning for patients with HCC (summarized in Table 2). Most prior studies correlated detection of mutations in particular genes such as TERT or BCL9 and RPS6KB1 with outcomes (OS and PFS) or advanced pathological features such as tumor size, PVTT, or TNM stage [34,70,85,86]. The quantitative analysis by mutation allelic fraction (MAF) in cfDNA testing also proved to be useful in some studies [31,44]. 

EM were less explored than mutations in HCC. EM were usually single-gene-based, such as LINE1, TFPI2, IGFB7, or small panels (APC, GSTP1, RASSF1A, and SFRP1) [59,87,88]. Interestingly, one study by Xu at al. constructed two major models based on methylation markers (CpG sites): one for diagnosis (*n* = 10 genes) and another for prognosis (*n* = 8). The former was also valuable in predicting tumor burden and poor outcomes [55].

**Table 2 ijms-24-14231-t002:** cfDNA mutations and epigenetic markers with prognostic value in hepatocellular carcinoma (HCC).

	Publication (n)	Methods	Tested Markers	Findings	Additional Information
Mutations	Oversoe et al., 2020 (95) [34]	Digital droplet PCR (ddPCR)	TERT (C228T)	*TERT* mutation conferred increased mortality (HR 2.16 (1.20–3.88), *p* = 0.01) when detected in plasma (adjusted HR 2.16 (1.20–3.88), *p* = 0.010), but not in tumor (adjusted HR 1.11 (0.35–3.56), *p* = 0.860).	In treatment-naïve HCC, mortality was more pronounced (HR 4.11 (1.73–9.76) *p*= 0.001) Detectable plasma TERT mutation was correlated with advanced TNM stage and vascular invasion (*p* < 0.0005) Non-concordance was associated with an early TNM stage.
	Jiao et al., 2018 (215) [85]	ddPCR	TERT (C228T and C250T)	Mutation-positive tumors with cirrhosis have worse prognosis (*p* = 0.0042). This association was significant in MVA	Mutations associated with worse OS (*p* = 0.0062) for all HCC patients (with and without cirrhosis). This association was not significant on MVA
	Yang et al., 2011 (60) [70]	Quantitative fluorescent polymerase chain reaction (FQ-PCR)	TERT	Plasma TERT DNA levels were closely related to tumor size (<5 vs. 5–10 vs. >10 cm), PVTT and TNM (I-II vs. III-IV) stage (*p* = 0.013, *p* = 0.010, and *p* = 0.029, respectively)	Cutoff–variable Survival analysis is not available.
	Youssef et al., 2023 (100) [86]	PCR	BCL9 and RPS6KB1	BCL9 gain and BCL9 + RPS6KB1 gain led to higher mortality rates and reduced survival times.	Prevalence of CNV gain in BCL9 and RPS6KB1 genes was detected in 14% and 24% of patients, respectively. Gain in both genes showed a high risk of HCC with elevated liver enzymes, Schistosomiasis, BCLC C, and PS > 1
	He et al., 2019 (29) [31].	Next Gen Sequencing (NGS)	2800 COSMIC hotspots in 50 high-frequency mutations in tumor genes.	MAF in the TP53, CTNNB1, PIK3CA, and CDKN2A increase the risk of larger tumors (>5 cm) MAF in the TP53, RET, FGFR3 and APC associated with multiple tumors or distant metastasis	35 mutant genes/50 tested were noted in tumors. TP53 was detected in 50% of tumors. Cut-off for MAF is >1%
	Kim et al., 2020 (107) [44]	NGS	2924 SNVs in 69 genes	MLH1 SNV, in combination with an increased ctDNA level (≥5.77 ng/mL), predicted poor overall survival among	OS was worse in HCC with high ctDNA (vs. low, *p* = 0.0014); High ctDNA with MLH1 mutation vs. high ctDNA + MLH1 witld type (WT) vs. low ctDNA + MLH1 WT
Epigenetic markers	Tangkijvanich et al., 2007 (85) [88]	Combined bisulfite restriction analysis PCR	Serum hypomethylation status of LINE-1 repetitive sequences	High serum LINE-1 hypomethylation associated with worse OS (10.5 vs. 5.5 m, *p* = 0.012)	High levels are associated with larger (>5 cm) and advanced (based on CLIP scoring). Cut off- 53.17 ± 7.74%,
	Huang et al., 2011 (72) [57]	MSRE-qPCR	APC, GSTP1, RASSF1A, and SFRP1	High levels of APC or RASSF1 methylation have worse outcomes. GSTP1 has a trend towards worst outcome (*p* = 0.062)	Methylated RASSF1A alone was independent risk factor for OS (hazard ratio = 3.262, 95% CI: 1.476–7.209, *p* = 0.003).
	Sun et al., 2013 (43) [59]	PCR	TFPI2 methylation percentage	Higher stage is associated with increased TFPI2 methylation percentage	Higher percentage with stage—39, 38, 64, 75, from Stages I–IV, respectively (percentage = methylated/unmethylated)
	Li et al., 2014 (136) [87]	PCR	IGFBP7	Methylation is associated with vascular invasion (84% versus 60%, *p* = 0.010).	Frequency of serum IGFBP7 promoter methylation in HCC patients was 65% (89/136)
	Xu et al., 2017 (1098) [55]	Deep sequencing of bis-DNA target-captured with molecular-inversion	BMPR1A PSD ARHGAP25 KLF3 PLAC8 ATXN1 Chr 6:170 Chr 6:3 ATAD2 Chr 8:20	Higher combined diagnostic score (cd-score) was associated with advanced TNM stage, poor response, early recurrence, and tumor burden	cd-score also has a diagnostic value.
	Xu et al., 2017 (1098) [55]	Deep sequencing of bis-DNA target-captured with molecular-inversion	SH3PXD2A C11orf9 Chr 17:78 PPFIA1 SERPINB5 NOTCH3 TMEM8B GRHL2	High combined prognosis score (cp-score) was associated with poor survival. Cp-score and TNM staging better than individual alone.	Cut-off for cp-score was >0.24. Cp-score was validated by MVA (HR: 2.405; 95%; *p* < 0.00)

HR—hazard ratio; OS—overall survival; BCLC—Barcelona Clinic Liver Cancer; MVA—macrovascualr invasion; PVTT—portal vein tumor thrombosis; MAF—Mutant allele frequency; PS—performance status; SNV—single nucleotide variants; CNV—gene copy number variants; MSRE—methylation-sensitive restriction enzymes-based quantitative.

## 4. Predictive Biomarkers

Genomic biomarkers have not been studied to predict the response to common LRTs performed for HCC such as TACE, TARE, SBRT, or systemic therapy (ICI and TKI). Very few studies have explored this area of HCC management and are summarized in Table 3. These studies follow the same trend as prognostic or POR studies. Most of these genomic biomarkers are non-specific (cfDNA levels), while some studies have used NGS or hotspot panels on pre-treatment samples. 

Sefriouri et. al studied the change in MAF of TERT mutations and cfDNA levels from baseline, post-TACE (day 2), and one month later [89]. They reported that patients who do not show any drop/change in either (TERT or cfDNA) are at an increased risk of having residual disease or progression. In another study, patients with TERT mutations who received TACE (44/130, rest had TKIs) had poorer OS (and large size) [90]. Higher cfDNA levels post-radiation (SBRT and external beam radiation) and TARE (from cfDNA testing on patients in SORAMIC trial before systemic therapy) were also indicators of poor response (Table 2) [91,92]. The latter study also gives us insight into biomarkers for sorafenib response as well. There were no studies that used cfDNA EM for this purpose. 

Nakatsuka et al. reported that higher baseline cfDNA levels were an independent negative risk factor among HCC patients receiving LRTs (TACE and RFA) and systemic therapy (TKIs and ramucirumab) [93]. Interestingly, a greater rise in cfDNA level post-therapy (day 2 of LRT and average of day 3 after systemic therapy) was an indicator of a favorable response. Higher baseline cfDNA levels and AFP (>400 ng/mL), along with TERT positivity, were associated with poor outcomes in patients treated with atezolizumab/bevacizumab combination. Patients with mutations in PIK3CA/mTOR pathway genes (PIK3CA, PTEN, TSC1, TSC2, and RPS6KA3) have poor response rates to TKIs, while mutations in WNT pathways had no effect on ICI response [29]. EM were not explored enough as predictive biomarkers for HCC. 

**Table 3 ijms-24-14231-t003:** cfDNA testing for predicting response to locoregional therapy and systemic therapy.

Publication	Methods	Tested Markers	Findings	Additional Information-Frequency
Sefrioui et al., 2022 (28) [89] TACE	ctDNA for TERT ddPCR cfDNA was quantified by the fluorometric method.	TERT hotspot cfDNA levels Samples collected prospectively: baseline (D − 1), day 2 (D + 2), and one month later (DM)	High-risk patients progressed in one month (80.0% vs. 4.3%, *p* = 0.001) and worse PFS (1.3 vs. 10.3 months, *p* = 0.002)	Score based on changes in cfDNA and ctDNA. High-risk or non-responders = no change in both; low-risk = change in at least one marker. cfDNA change from D-1 to D M+ 1 of > 31.4% (44.4% vs. 3.8%) and ctDNA change >0% as high risk (50.0% vs. 5.0%)
Park et al., 2018 (55, 34 fractionated XRT and 21 SBRT) [91]	Ultraviolet-visible spectrophotometer	cfDNA levels before and after XRT	High pre-XRT cfDNA levels are associated with advanced disease (*p* = 0.049) and larger tumors (*p* = 0.017). High post-XRT cfDNA level is associated with poor response rates (*p* = 0.017), local control (*p* = 0.006), or intrahepatic recurrence rates (*p* = 0.035)	Cut-off values for cfDNA levels: 33.65 ng/mL before XRT is and 37.25 ng/mL after CRT)
Hirai et al., 2021 (130, 44 TACE and 86 with lenvatinib or sorafenib) [90]	Digital droplet- polymerase chain reaction (dd-PCR)	TERT	Detection of TERT mutation and high fractional abundances (≥1%) associated with worse OS (*p*< 0.01).	TERT promoter mutations correlated with large intrahepatic tumor size (*p* = 0.05) and high des-gamma carboxyprothrombin (*p* = 0.005)
Alunni-Fabbroni et al., 2019 (13, 10, TARE + sorafenib, 2 RFA + sorafenib, 1placebo) [92]	Next-generation sequencing (NGS)	597 gene panel. Samples were collected between local therapy and beginning of sorafenib-based systemic therapy (T1), then at 3 time points—8 weeks apart from the date of starting sorafenib (T2, T3, T4)	High cfDNA concentrations at T1 associated with metastasis (*p* = 0.012) CYP2B6 variant detection at T1–T2 associated with worse OS (*p* = 0.013)	High T2 has a trend towards significance for metastasis (*p* = 0.07) T1 and T2 concentration do not affect OS; however, high levels at T3 (*p* = 0.057) and T4 (*p* = 0.095) have trends. Early detection of BAX (for MVI, *p* = 0.014) and HFN1A (for liver cirrhosis, *p* = 0.032) confer worse clinicopathological features.
Oh et al., 2019 (151) [94] Sorafenib	Whole genome sequencing Baseline	cfDNA level, VEGFA-to-EIF2C1 ratios I score for genomic instability.	Patients who failed sorafenib had significantly higher cfDNA levels (0.82 vs. 0.63 ng/μL; *p* = 0.006) and I-scores (3405 vs. 1024; *p* = 0.0017) than those achieving disease control. VEGFA: EIF2C1 ratio was not significantly associated with treatment outcomes.	94% had previous therapy cfDNA-high group had a worse time to progression compared to the low group (2.2 vs. 4.1 months; HR = 1.71; *p* = 0.002) and OS (4.1 vs. 14.8 months; HR = 3.50; *p* < 0.0001) Genome instability-high group progressed earlier (2.2 vs. 4.1 months; HR = 2.09; *p* < 0.0001) and had worse OS (4.6 vs. 14.8 months; HR = 3.35; *p* < 0.0001).
Fuji et al., 2021 (24) [30] Lenvatinib	NGS cfDNA at the baseline and 4 weeks later	Guardant 360 panel	Reduction MAFmean 4 weeks after treatment was useful in predicting PR and CR, but not for SD or PD	Baseline mutations and MAF change were not indicative of OS or PFS MAFmean change ≥ 0 vs. <0 HR 8.4, 95% CI = 2.3–31.2, *p* = 0.002) as an independent factor was associated with poor PFS. MAFmean reduction was more sensitive than AFP
Fu et al., 2022 [79] (*n* = 258, lenvatinib + ICI)	PCR	High-risk genes APC, ARID1A, CDKN2A, FAT1, LRP1B, MAP3K1, PREX2, TERT, and TP53	Preserved FAT1 or LRP1B variants without TP53 associated with poor PFS (HR = 17.1, *p* < 0.001).	Samples were collected post-surgery
Nakatsuka et al., 2021 (100 pre and 87 post treatment) [93] Treatments received include TACE, RFA (day 2), TKIs (lenvatinib, 27; sorafenib, 6; regorafenib, 1), and 1 ramucirumab *	Targeted ultra-deep sequencing	22,0009 coverage in a panel of 275 cancer-related genes	Greater increase in the post-treatment cfDNA levels (>66.5 ng/mL) associated with increased response (AUC of 0.807, sensitivity 60.0%, specificity 92.9%)	In particular, the detection rate increased significantly from 31 to 54% in the systemic-therapy-treated cases (*p* = 0.045). A higher baseline cfDNA (>70.7 ng/mL) was associated with poor survival (5.5 m vs. 13.7 m, *p* < 0.001) and is an independent factor for OS. In Lenvatinib responders, AMER1, MLL3, and NOTCH2 were mutated.
Matsumae et al., 2022 (85) [95] (Atezolizumab/bevacizumab)	.	Custom-made panel for detecting mutations in 25 HCC-related cancer genes	Higher cfDNA levels have poor objective response rates (PR and CR, *p* = 0.03) and PFS (*p* = 0.021) TERT-positive patients have poor OS (*p* = 0.001), but no difference in PFS.	Higher plasma cfDNA (≥2.23 ng/µL), TERT positivity, high AFP (>400 ng/mL) were independent prognostic factors for OS OS–TERT+ and AFP− high < TERT+ or AFP− high < TERT− and AFP low (*p* < 0.001)
Von Feleden et al., 2021 (51, TKI = 23; ICI = 38) [29]	Ultra-deep sequencing for 25 genes ddPCR for TERT	PI3K/mTOR pathway genes for patients on TKI WNT pathway genes for patients on ICI	Mutation-positive patients have lower PFS (2.1 vs. 3.7 months, *p* < 0.001) No effect of mutations in WNT pathway on ICI response	Serial testing–not clear if it helps in predicting response In some patients’ primary resistance, there is no change in mutations profiles, but there is increase in VA. In some patients with partial response, mutations have disappeared—not clear if they help in monitoring response

TACE—Transarterial chemo embolization; PFS—progression-free survival; OS—overall survival; XRT—radiation; SBRT—stereotactic radiation therapy; HR—hazards ratio; MVI—macrovascular invasion; MAF—mutation allelic fraction; TKI—tyrosine kinase inhibitor; ICI—immune-checkpoint inhibitor; * post treatment samples were collected on day 2 for TACE and RFA (radiofrequency ablation), and average of day 3 for systemic therapy.

## 5. Conclusions

HCC is rising in incidence and overall outcomes remain poor. Despite having robust screening procedures in place, patients are often diagnosed in advanced stages with limited effective treatment options. A lack of reliable biomarkers that could improve current POR and predict or monitor the response to LRT and ST presents a clinical challenge for patients and treating physicians alike. One promising area of interest involves the use of cfDNA testing; however, serial improvements are needed before widespread use. Early reports of EM are promising and should be further explored on how to best incorporate it in current practice (summarized in Figure 1 below). Non-specific cfDNA level testing can help in monitoring treatment response or assessing the tumor burden. Its value without specific mutation or EM testing is questionable in POR. Developing models with specific mutations and EM in blood with high detectable rates will provide a non-invasive biomarker tool for risk-stratification perioperatively, estimate the survival, be vigilant of advanced pathological features (tumor size, PVTT), and help with treatment selection resistance monitoring. cfDNA testing to assist treatment selection (ST vs. LRT) in non-metastatic advanced and intermediate stage HCC and monitoring the treatment response could significantly impact outcomes in HCC patients. 

## Figures and Tables

**Figure 1 ijms-24-14231-f001:**
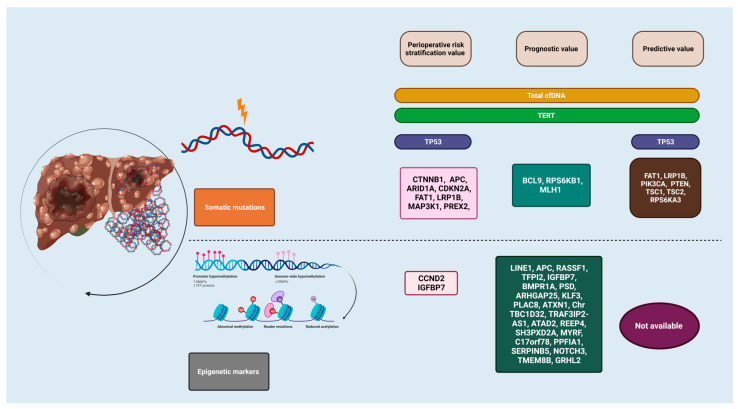
Current status of cell-free DNA testing in hepatocellular carcinoma.

## Data Availability

Not applicable.

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
