# Peer review of "Is Cell-Free DNA Testing in Hepatocellular Carcinoma Ready for Prime Time?"

_ijms, 2023, doi:10.3390/ijms241814231_

Round 1

Reviewer 1 Report

The authors present a narrative review focusing on Cell-Free DNA testing in patients with HCC. The title reflects the subject of the manuscript. It presents a clear and clinically useful message. It is well written in terms of clarity, style, and use of English language. The discussion section is sufficiently detailed and explains adequately the purpose of this study in the context of published information. The conclusion accurately and clearly explains the main result. The length of the manuscript is ideal. All references are appropriate and current.

Minor points

- The aim of the article needs to be highlighted in the abstract and ending of the introduction section

Author Response

  1. The aim of the article needs to be highlighted in the abstract and ending of the introduction section
  • Reply by the authors: We appreciate the feedback. We added the following lines to the abstract, We highlighted the unmet needs in the HCC management and the current role of cfDNA testing in HCC in addressing them, and following lines to the end of the introduction (lines 45-47), In this review, we focused on the gaps in HCC management and summarized the current status of cfDNA testing in addressing them.

Reviewer 2 Report

In this manuscript, Jeepalyam et al. described the current utility of cell-free DNA testing in hepatocellular carcinoma. This manuscript covers up-to-date information. This reviewer has a few comments.

Major comments: 

1. The biomarker for HCC may differ according to the cause of cirrhosis. It should be discussed.

2. Among the three applications of cell-free DNA testing, such as POR, prognostic value, and predictive biomarkers, the advantages, and disadvantages for each cell-free DNA level, mutations, and epigenetic markers should be discussed.

Major comments: 

1. In tables, the year number should be added after the author’s name.

2. “Nakatuska” on page 10 of 23, line 193, and page 13 of 23 should be “Nakatsuka”.

Author Response

  1. The biomarker for HCC may differ according to the cause of cirrhosis. It should be discussed.
    • Reply by the authors: We appreciate the feedback. We added the following lines to section 1.2 (lines 131-138), Etiology-specific cfDNA markers were reported in previous studies that can help in certain situations. The hepatitis B (HBV) carriers with HCC tend to have higher circulating ERBB2 and TERT mutations, higher methylation rates in RASSF1, TFPI2, TRG5 (along with AFP), and XPO4, and low methylation rates in CDKN2A than those without HCC [43,59,63,70-72]. Higher RASSF1 methylation rates are frequent in hepatitis C (HCV) pa-tients with HCC (compared to HCC-negative) [73]. Some of them can help in detecting the recurrence (e.g., virus‐host chimera DNA (vh‐DNA), generated from junctions of HBV integration in the HCC chromosome in HBV-HCC patients) or forecasting the outcomes (e.g., higher cfDNA levels in HCV-carriers) [74,75].
  1. Among the three applications of cell-free DNA testing, such as POR, prognostic value, and predictive biomarkers, the advantages, and disadvantages for each cell-free DNA level, mutations, and epigenetic markers should be discussed.
    • Reply by the authors: We appreciate the feedback. We addressed this specific question in the following lines of the conclusion section (lines 228-234), Non-specific cfDNA level testing can help in monitoring treatment response or assessing the tumor burden. Its value without specific mutation or EM testing is questionable in POR. Developing models with specific mutations and EM in blood with high detectable rates will provide a non-invasive biomarker tool for risk-stratification perioperatively, estimate the survival, be vigilant of advanced pathological features (tumor size, PVTT), and help with treatment selection resistance monitoring.
  1. In tables, the year number should be added after the author’s name.
      1. Reply by the authors: We appreciate the feedback. We added the years to the tables.
  2. “Nakatuska” on page 10 of 23, line 193, and page 13 of 23 should be “Nakatsuka”.
      1. Reply by the authors: We appreciate the feedback. We addressed the spelling error.

Reviewer 3 Report

The manuscript entitled " Is Cell-Free DNA Testing in Hepatocellular Carcinoma Ready for Prime Time?" has been reviewed. This paper is well written.  However, some questions need to be asked

This paper focuses on cell-free DNA (cfDNA) and aims to aid in the diagnosis or treatment of HCC.

The authors have found the epigenetic marker to be useful in cfDNA.

Perhaps cfDNA is only measured when HCC is detected, but is there a link between the size of the HCC and the test results? Or is it related to the histological type of HCC? Does the degree of cirrhosis have an effect on cfDNA?

The authors conclude that cfDNA may be useful for monitoring treatment in non-metastatic advanced and intermediate stage HCC, but is it of value in metastatic cases? HCC is predominantly intrahepatic metastasis,

none

Author Response

  1. Perhaps cfDNA is only measured when HCC is detected, but is there a link between the size of the HCC and the test results? Or is it related to the histological type of HCC? Does the degree of cirrhosis have an effect on cfDNA?
    • Reply by the authors:
      • In our literature review, we did come across many studies where cfDNA was detected in non-HCC patients, but the level was higher in HCC-patients. As the focus of our paper was on the management of HCC we did not elaborate on those patients where diagnosis was the focus. We now added lines in the introduction as a response to other reviewers that address this.
      • Studies did report a correlation between size and other characteristics of HCC with some cfDNA markers mentioned in the text and even in relevant studies in the tables (additional information columns).
      • The degree of cirrhosis and cfDNA is an interesting question. To our knowledge, there are no studies that reported this specific question.
  1. The authors conclude that cfDNA may be useful for monitoring treatment in non-metastatic advanced and intermediate stage HCC, but is it of value in metastatic cases? HCC is predominantly intrahepatic metastasis
    • Reply by the authors: cfDNA testing has the potential to be a valuable tool in metastatic patients. Unfortunately, we do not have enough studies focusing on it. We believe the focus on those patients must be on the predictive value as it may guide in treatment selection (local therapy with systemic therapy). The reported ones (Table 3) could not give clinically helpful information. The goal of this paper was to highlight the need (addressed in section 1.1) and encourage studies in this area.